# Social media and public perception as core aspect of public health: The cautionary case of @realdonaldtrump and COVID-19

**Agustín Fuentes**[1]*, **Jeffrey V. Peterson**[2]

**1** Department of Anthropology, Princeton University, Princeton, NJ, United States of America, **2** Department of Anthropology, University of Notre Dame, Notre Dame, IN, United States of America

☯ These authors contributed equally to this work.
* afuentes2@princeton.edu

## Abstract

The social media milieu in which we are enmeshed has substantive impacts on our beliefs and perceptions. Recent work has established that this can play a role in influencing understanding of, and reactions to, public health information. Twitter, in particular, appears to play a substantive role in the public health information ecosystem. From July 25th, 2020 to November 15th, 2020, we collected weekly tweets related to COVID19 keywords and assessed their networks, patterns and properties. Our analyses revealed the dominance of a handful of individual accounts as central structuring agents in the networks of tens of thousands of tweets and retweets, and thus millions of views, related to specific COVID19 keywords. These few individual accounts and the content of their tweets, mentions, and retweets are substantially overrepresented in terms of public exposure to, and thus interaction with, critical elements of public health information in the pandemic. Here we report on one particularly striking aspect of our dataset: the prominent position of @realdonaldtrump in Twitter networks related to four key terms of the COVID19 pandemic in 2020.

## Introduction

The social media milieu in which we are enmeshed has substantive impacts on our beliefs, actions and reactions to information, and thus is a relevant arena for understanding public health. Digital media helps shape cultural beliefs and practices [1]. In particular, social media is a major factor in the exposure to information and the structuring of social/cultural landscapes in the contemporary world [2]. As humans receive social/cultural information, we create mental representations and neural connections, developing ideas, understandings and beliefs that become neurobiological, social, and psychological processes influencing how we see and experience the world, with particularly popular, socially pervasive and central cultural figures/concepts having disproportionate influence [3–5]. Social media exposure and engagement influences the tone, tenor and content of public perceptions, and action, in a variety of arenas (e.g. politics [6]), and there is substantive evidence that social media is intricately involved with health [7].

**Data Availability Statement:** The data underlying this study are available on Dryad (https://doi.org/10.5061/dryad.69p8cz922).

**Funding:** This project was funded by a donation to Princeton University and A. Fuentes from the John Templeton Foundation (https://www.templeton. org/) The funders had no role in study design, data collection and analysis, decision to publish, or preparation of the manuscript.

**Competing interests:** The authors have declared that no competing interests exist.

There is widespread agreement that due to its influence, extensive use and the frequency and ease of access by the general public, social media can and should be used for beneficial dissemination of public health information, especially in the context of the COVID19 pandemic [8, 9]. There is also evidence that world leaders' social media use, especially Twitter, has been substantive and influential during the COVID19 pandemic [10]. As Rufai and Bunce [11] note "Twitter may represent a powerful tool for world leaders to rapidly communicate public health information with citizens." However, they urge "general caution when using Twitter for health information" with a "preference for tweets containing official government-based information sources."

Our analyses reveal the dominance of a few individual accounts as central actors in the networks of tens of thousands of tweets on the topic. These few individual accounts and the content of their tweets, mentions, and retweets are thus substantially overrepresented in terms of public exposure to, understanding of, and reaction to, critical elements of public health information during the pandemic [12]. Most concerning is the outcome we report here: the prominent presence of @realdonaldtrump as one of the central structuring figures in the global Twitter discourse related to the COVID19 pandemic in 2020.

There is good evidence that Donald Trump intentionally used Twitter in a manner so as to structure the power and content of "Presidential talk" into a new mode of control and influence [13, 14]. A recent study demonstrated that Donald Trump specifically exploited Twitter as a tool to divert/direct the attention of mainstream media in relation to his personal goals rather than public good [15]. There is also substantial evidence documenting Donald Trump's antagonism towards scientific data and public health assessments/recommendations and the officials/researchers who generated them [16] and that such antagonism has had deleterious impacts on public health and health systems in the USA [17]. There is also evidence of the negative and harmful impact Donald Trump had as a Twitter user, resulting in his being suspended permanently from the social media platform on January 8th, 2021 (see Twitter's statement on the permanent suspension [18]). Finally, there is also supporting evidence that Donald Trump's activity on Twitter significantly affected content and patterns of discourse on social media writ large, such as a 73% decline in online misinformation about election fraud in the week after Twitter permanently suspended @realdonaldtrump [19].

We contend that @realdonaldtrump played a particularly prominent and harmful role in damaging public health efforts in the USA during the COVID19 pandemic of 2020 (and beyond) via the Twitter platform. Here we offer an analysis supporting this contention and suggest reasons why reconciling with this potential for information bias is critically important for future public health crises. Our goal here is to offer a retrospective look at the potential impact of a single, potentially malicious, individual Twitter account on public health discourse as fodder for future considerations of the management and structuring of social media content in public health crises.

## Materials and methods

From May to July 2020 our research group conducted a preliminary examination of six mainstream news media websites, multiple state public health websites and articles/essays by six prominent science writers as initial research for a larger project to examine communication of scientific and public health information during the COVID19 pandemic. Drawing on the preliminary data from May through July we identified seven keywords to sample narratives about the COVID19 pandemic on Twitter: 1) Coronavirus origin; 2) Coronavirus vaccine; 3) COVID19; 4) Fauci; 5) Mask; 6) Open (school OR economy); and 7) Social distancing. From July 25th, 2020 to November 15th, 2020, using NodeXL Pro we made weekly collections of

tweets that include these individual keywords. NodeXL Pro software accesses the Application Programming Interface (API) of Twitter to compile tweets within specified parameters (e.g., keywords). We instructed NodeXL to pull relevant tweets for each of our keywords separately, allowing us to compile unique keyword Twitter networks for each topic. Because this sampling method does not include every tweet using each keyword, our subsequent network analysis involves partial networks. Each of our keyword networks contains a maximum of 10,000 tweets to keep network download times and file sizes manageable while also collecting a representative sample of tweets using the identified keywords. The period of July 25th to November 15th was selected so that the Twitter project data coincided with concurrent elements of the larger project on tracking COVID19 information.

We also used NodeXL Pro software for social network construction and analysis. For each keyword network we calculated the total nodes, edges, and mean geodesic distance. We further assessed in-degree and betweenness centrality measures for every node in each network. In-degree centrality in the Twitter network is measured by mentions, replies, tweets, retweets, and mentions in retweets registered to an account in the given network. High in-degree centrality for a given Twitter account indicates that a high number of Twitter users engaged with it in the respective keyword network. Betweenness centrality in the Twitter network is the number of times a given node appears on the shortest path between two other nodes. Therefore, high betweenness centrality for a given Twitter account indicates that they are a crucial node for information transmission between other Twitter users. We then identified the top ten Twitter accounts in each weekly keyword network for both centrality measures. The accounts appearing in the top ten for in-degree and betweenness centrality in three or more weekly networks for each keyword network series were identified as key contributors to the Twitter discourse surrounding those topics. We calculated the mean centrality of each account across all weeks they appeared in a keyword network's top ten in-degree and betweenness centrality values. We then calculated a metric to assess the relative impact of each account across keyword network series by multiplying their mean in-degree and betweenness centrality by the ratio of weeks they appear in that keyword network's top ten accounts (by centrality) per total number of networks in that keyword series. We call this measure the Adjusted Influence Centrality (AIC). This metric is important given that in large participant pools some individuals may miss some events. As such, comparing the mean centrality measures of each individual for the events at which they are present is insufficient as one should scale those means to reflect the relative frequency of each node's presence at the successive events. Thus, the AIC is a useful metric for determining which nodes sustain influence over successive partial networks when the participant pool may vary. In this analysis, the AIC allows us to measure which Twitter accounts have a sustained influence over Twitter keyword networks across the 17 weeks of data collection. For instance, an account with a moderate average centrality measurement after appearing in 11 of the 17 weekly networks may have a greater influence on the keyword topic discourse overall than an account appearing in only three of the 17 weekly networks, even if that latter account peaks to the highest recorded centrality for that keyword network series in one of those weeks. Therefore, the AIC assesses consistency of impact for specific Twitter accounts across multiple keyword networks in a series.

Our research was conducted under the permission from the Princeton University Institutional Review Board Protocol IRB# 13070, approved 17 July 2020 and following the Internet Research Ethical Guidelines 3.0 from the Association of Internet Researchers. The data set used for the analysis we present here is openly available on DRYAD (DOI: https://doi.org/10.5061/dryad.69p8cz922)

## Results and discussion

While we found a number of different twitter accounts scored high in one or more of our network measures in different weeks across the entire survey period, @realdonaldtrump was the top or in the top two in in-degree and betweenness centrality, and AIC, across the entire data collection period for the following four keywords: "Fauci," "Mask," "Open (school or economy)," and "Social distancing," with one exception (Table 1). Also, @realdonaldtrump made more appearances per week in the top ten accounts based on centrality for these networks than any other single Twitter account in each of these keyword networks across the entire data collection period.

Between July 25 and November 15, 2020, @realdonaldtrump received a combination of more mentions, replies, tweets, retweets, and mentions in retweets (high in-degree centrality) in two of the four keywords than any other account, the specific pattern of which varied depending on the keyword network. With the highest betweenness centralities across the entire dataset @realdonaldtrump was the peak common node for information transmission between other Twitter users for these four keywords. Finally, @realdonaldtrump had the highest AIC measure in every case except for the "Mask" keyword network series' in-degree centrality AIC. This one exception to top rank was due to the massive surge of @joebiden in the "Mask" keyword network between October 10[th] and November 1[st], 2020 (immediately preceding the presidential election in the USA). Otherwise, @realdonaldtrump was between one and fourteen times higher in AIC value than the nearest other Twitter individual across the four categories, meaning that the consistency of impact of @realdonaldtrump across multiple keyword networks in a series was substantially dominant.

@realdonaldtrump was the central node in the Twitter landscape of "Fauci," "Mask," "Open (school or economy)," and "Social Distancing" between July 25 and November 15, 2020. In that time period the USA went from ~65,000 infections per day to ~136,000 infections per day, and from 147,000 deaths to 246,000 deaths from COVID19 [20]. There is no debate that Donald Trump's use of Twitter was both highly influential and filled with misinformation. In an eight-year study of Trump's Twitter content Pain and Chen [21] found that:

> "His tweets reach Americans directly and give him an enviable method of talking directly to the people. Thus, they could be a boon of deliberative dialogue. Our analysis shows the opposite. His tweets perpetuate division, misinformation, and lies and lack any semblance of deliberative discourse. He does not use evidence to support points, which could further his deliberative reach, and instead, he attacks in a manner that leads to greater divisiveness and rancor." (pg. 8)

As noted earlier, there is robust documentation of Donald Trump's antagonism towards health/science research and officials [16], his use of Twitter to divert media focus [13, 14], and

**Table 1. Ranking for @realdonaldtrump in select keyword networks.**

|  | @realdonaldtrump Ranking | | | |
|---|---|---|---|---|
|  | Peak in-degree centrality across keyword network series | AIC for in-degree centrality across keyword network series | Peak betweenness centrality across keyword network series | AIC for betweenness centrality across keyword network series |
| Fauci | 2[nd] | 1[st] | 2[nd] | 1[st] |
| Mask | 4[th] | 2[nd] | 2[nd] | 1[st] |
| Open (school or economy) | 1[st] | 1[st] | 1[st] | 1[st] |
| Social distancing | 1[st] | 1[st] | 1[st] | 1[st] |

the central role of Twitter in the potential for disinformation regarding COVID19 [9]. It would appear from both established long-term general patterns and from specific tweet content [16, 22] that @realdonaldtrump tweeted disinformation and harmful information regarding COVID19. Thus, given that he had over 88 million followers receiving his tweets, we can argue with reasonably certainty that tweeting from @realdonaldtrump potentially harmed public health understanding and engagement with COVID19. But that is not the only context in which @realdonaldtrump influenced Twitter discourse on this topic. Given the structure of our data collection and analyses we are measuring not only content originated or retweeted by @realdonaldtrump but also content generated by those tweeting "for" or "against" him and/or criticizing him and/or mentioning him and/or retweeting him (e.g. our AIC measures). In these contexts, even those of rebuttal or critique, the presence of @realdonaldtrump as a central node structuring the Twitter discourse guarantees broadscale exposure of the content from @realdonaldtrump and those who intentionally support it. These data and analyses support the assertions that between July 25 and November 15, 2020, (and likely before and after those dates until the account's suspension) a substantive component of the public social informational milieu of the COVID19 pandemic was shaped and amplified via @realdonaldtrump.

Given this documented overall central role in twitter networks, what might the dominance of @realdonaldtrump in Twitter discourse surrounding the terms "Fauci," "Mask," "Open (school or economy)," and "Social Distancing" mean for the public's understanding of, and actions towards, specific and serious public health issues related the COVID19 pandemic?

Three of these four terms ("Mask", "Social Distancing", and "Open (school or economy)") are directly linked to public health actions clearly and consistently stated and reinforced by a majority of State public health departments, by the Centers for Disease Control (CDC) and the White House Coronavirus Task Force. The recognized need for masks and social distancing as key prevention measures was ubiquitous across the entirety of the COVID19 research and public health landscape by April 2020. As of July 27, 2020, mask mandates were officially required by a majority of US States and emphatically emphasized by the CDC [23]. By September 2020, the White House Coronavirus Task Force and nearly all States had and supported mask mandates. Yet, a large percentage of the USA population resisted wearing masks, increasing transmission risk. Across the time period of our data collection activity by @realdonaldtrump and public statements and actions by Donald Trump were largely negative on mask wearing (including mocking presidential candidate Joe Biden repeatedly for wearing one). @realdonaldtrump did tweet one significant positive mask statement along with an image of wearing a mask on July 20[th], 2020, but shortly thereafter returned to the pattern of negative "mask" contents (in tweets and actions). While the data are only correlational, we argue that the central presence of @realdonaldtrump in appearances of the term "Mask" on Twitter is likely to have influenced public perception of their use, the risk of not using them, and of actions by the public. This assertion is supported via the recent finding that "self-reported mask-wearing increased separately from government mask mandates, suggesting that supplemental public health interventions are needed to maximize adoption and help to curb the ongoing epidemic" [24]. One final bit of support for the pattern we propose is that nearly all states, and all federal health agencies, recommended if not required masks by September 2020, yet mask use in the USA dropped slightly from ~70% to ~65% between the end of July 2020 and early October 2020, when it began to rise to the level of ~75% (by later November 2020 [25]). Interestingly, this time frame for the rise in mask usage maps to the period from early October when Donald Trump contracted COVID19 and to that of between October 10 and November 1[st] when @joebiden replaced @realdonaldtrump as the most prominent twitter user in in-degree centrality for the term "Mask."

The case with "Social distancing" is similar, however public patterns and changes in social distancing behavior are more difficult than that of mask wearing to effectively assess in direct connection to pattern of @realdonaldtrump tweets. The case for "Open (school or economy)" is also similar to "Mask" in that @realdonaldtrump consistently tweeted against restrictions on gatherings, hospitality, school closures and related issues, and the "Open" controversy remained prominent as a public health concern in the USA during our data collection period. Regardless of whether or not the Twitter users were supportive of the tweets and retweets of @realdonaldtrump, arguing against them, or not responding directly to them, many (or most) of the tens of millions of individuals who saw tweets with these keywords in them, were exposed to the structuring of the conversation around activity by @realdonaldtrump.

Finally, the result of @realdonaldtrump being so dominant on Twitter networks involving the keyword "Fauci" is centrally concerning as Dr. Fauci's views most often reflected the up to date scientific and public health information. This pattern implies that the context in which most individuals who engage with Twitter saw tweets associated with the term "Fauci" were likely influenced or shaped by the tweets/retweets of @realdonaldtrump. We suggest that this pattern, as with the other three key terms, created a mode of influence on the public that corresponded with continued high-risk behavior, derision of valid scientific and public health information, and subsequent growing infection rates, setting the stage for the massive increases in infections of November and December of 2020, and January of 2021.

## Conclusion

Our assessment of Twitter patterns related to the COVID19 pandemic offers a cautionary tale for public health. The Lancet Commission on Public Policy and Health in the Trump Era clearly document the specific damage done to USA public health policy, and public health itself, by the Trump administration [17]. The report lays out a number of specific executive and legislative actions to remediate such harms, but does not address social media. We, along with many others [9–14], suggest adding increased analytic and scientific attention to social media as a platform, and process, for the structuring of public health outcomes. We recognize that the data set provided here is only correlational, but given the numerous research findings regarding the use of social media by Donald Trump, and his administration's well documented negative impact on public health [17], we argue that our data and analyses offer support to the assertion of a substantive influence of @realdonaldtrump in negatively shaping public health perceptions and thus outcomes.

We add our voice to the contention that there should be careful analyses and monitoring (and intervention?) of public health disinformation and harmful assertions on social media [8, 9, 26]. Twitter's removal of @realdonaldtrump following the violent mob attack on the USA Capitol building January 6[th], 2021 is an extreme example, but one might also note that the actions of @realdonaldtrump between July 25 and November 15 in regard to the COVID19 pandemic may have been as dangerous and intentionally misleading but much more lethal than the lead up to the January 6[th] mob event.

This fundamental centrality of @realdonaldtrump in public health related discourse on Twitter begs the question of why many analyses of public health activity, both positive and negative, occurred during the pandemic in 2020 without mentioning and directly confronting the activity of @realdonaldtrump. While there have been some such assessments [12, 16, 26] much of the Twitter analyses research has focused on the veracity of information being distributed on Twitter [27, 28]. Recent social science investigation has focused in on the role of political actors on Twitter [11, 15, 29, 30] but this area of study has yet to develop explicit linkages with public health related behavior and outcomes. This may be due to current approaches

viewing the patterns on Twitter as a phenomenon primarily about information dissemination and not yet making the links to the effects on specific patterns of public health-related behavior influenced by information consumption of Twitter users. One reason for this absence of linkage may be that there has been little in the way of responses/challenges to these patterns on Twitter by the medical/public health community (but see [8, 9]).

Given the increasing evidence that social influence is a significant force in shaping individuals' responses to pandemic contexts [31] and the centrality of integrative social scientific approaches to public health in this and future pandemics [32] our results lead us to ask the question: Can the activities of individual social media accounts be seen as analogous to the patterns of infectious biological or noxious environmental agents in the assessment and management of public health threats? We think they might, but do not have sufficient data to accurately assess the possibility at present. Our data do support the assertion that investigating social media social network processes as they influence public health related behavior and outcomes might offer insight into public perception and reaction in the face of pandemic and other health crises. Overall, we suggest that public health research and assessments can benefit from expanding engagement with real-time social media structures and activities that may act like biological agents to influence, and likely shape, the perceptions, concepts, beliefs, and actions of the broader public.

## Acknowledgments

We thank two anonymous reviewers for their substantive and beneficial critiques of earlier version of this manuscript. We also thank members of the 2020 broader project research team (Hilcia Acevedo, Mary Devellis, Doruntina Fida, Jennifer Lee, Alexandra Marino, and Kevin Ramos) for their contributions to affiliated datasets.

## Author Contributions

**Conceptualization:** Agustín Fuentes.

**Data curation:** Jeffrey V. Peterson.

**Formal analysis:** Agustín Fuentes, Jeffrey V. Peterson.

**Investigation:** Agustín Fuentes, Jeffrey V. Peterson.

**Methodology:** Agustín Fuentes, Jeffrey V. Peterson.

**Project administration:** Agustín Fuentes.

**Resources:** Agustín Fuentes.

**Software:** Jeffrey V. Peterson.

**Supervision:** Agustín Fuentes.

**Writing – original draft:** Agustín Fuentes, Jeffrey V. Peterson.

**Writing – review & editing:** Agustín Fuentes, Jeffrey V. Peterson.

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
