## [Decision Letter · Decision Letter 0]

12 Apr 2021

PONE-D-21-05624

Social media and public perception as core aspect of public health: the cautionary case of @realdonaldtrump and COVD19

PLOS ONE

Dear Dr. Fuentes,

Thank you for submitting your manuscript to PLOS ONE. After careful consideration, we feel that it has merit but does not fully meet PLOS ONE’s publication criteria as it currently stands. Therefore, we invite you to submit a revised version of the manuscript that addresses the points raised during the review process.

The reviewers suggest some minor revisions that you are encouraged to consider and respond to.

We look forward to receiving your revised manuscript.

Kind regards,

Siân E Halcrow, Ph.D.

Academic Editor

PLOS ONE

Journal Requirements:

Reviewers' comments:

Reviewer's Responses to Questions

**Comments to the Author**

1. Is the manuscript technically sound, and do the data support the conclusions?

Reviewer #1: Yes

Reviewer #2: Yes

2. Has the statistical analysis been performed appropriately and rigorously? 

Reviewer #1: Yes

Reviewer #2: Yes

3. Have the authors made all data underlying the findings in their manuscript fully available?

Reviewer #1: Yes

Reviewer #2: Yes

4. Is the manuscript presented in an intelligible fashion and written in standard English?

Reviewer #1: Yes

Reviewer #2: Yes

5. Review Comments to the Author

Reviewer #1: Abstract:

No concerns

Introduction:

Paragraph 3, line 98 is really something that should be in the discussion and not in the intro.

Methods:

Why were those specific 7 words chosen. Can you please explain why these were chosen as opposed to pandemic/lockdown/etc?

Has the AIC been previously derived or is this the creation of the authors. It’s logical but this is this measure truly valid?

Line 166 there is a random “s” there.

Discussion:

Line 243 – I would use another word other than “ridiculing”. Although likely his intention its presence is more of an opinion than staying with the facts.

Conclusion:

With your questions in the last paragraph, can you suggest some answers/solutions/future directions for research? How can we answer those questions before intervening as suggested.

Reviewer #2: The manuscript addresses an important and relevant topic. It is well-done research. However, I would recommend some minor improvements before publication.

-In the manuscript, there should be more relevant references. In total, there are only 26 references including non-academic sources. The COVID-19 pandemic has been quite studied in social science research. Therefore, authors should add some references to strengthen their manuscript. For example, these articles are closely related to the manuscript:

Casero-Ripollés, Andreu (2020).“Impact of Covid-19 on the media system. Communicative and democratic consequences of news consumption during the outbreak”. El profesional de la información, 29(2), e290223.

Haman, M. (2020). The use of Twitter by state leaders and its impact on the public during the COVID-19 pandemic. Heliyon, 6(11), e05540. (you cite a pre-print)

Yum, S. (2020). Social network analysis for coronavirus (COVID‐19) in the United States. Social Science Quarterly, 101(4), 1642-1647.

Pulido, C. M., Villarejo-Carballido, B., Redondo-Sama, G., & Gómez, A. (2020). COVID-19 infodemic: More retweets for science-based information on coronavirus than for false information. International Sociology, 35(4), 377-392.

Rodríguez, C. P., Carballido, B. V., Redondo-Sama, G., Guo, M., Ramis, M., & Flecha, R. (2020). False news around COVID-19 circulated less on Sina Weibo than on Twitter. How to overcome false information?. International and Multidisciplinary Journal of Social Sciences, 9(2), 107-128.

Haman, M. (2021). Twitter Followers of Canadian Political and Health Authorities during the COVID-19 Pandemic: What Are Their Activity and Interests? Canadian Journal of Political Science, 54(1), 134–149.

I would also recommend mentioning research regarding adoption/using Twitter by political actors. In this way, the authors can emphasize the importance of social science research on Twitter. For example:

Barberá, P., & Zeitzoff, T. (2018). The New Public Address System: Why Do World Leaders Adopt Social Media? International Studies Quarterly, 62(1), 121–130.

Haman, M., Školník, M. (2021). “Politicians on Social Media. The online database of members of national parliaments on Twitter”. Profesional de la información, 30(2), e300217. https://doi.org/10.3145/epi.2021.mar.17

-Regarding methods - The methods are well developed. However, the authors should explain why they chose the period from "July 25th, 2020 to November 15th, 2020" and not a different one. Is it somehow connected with the US general election?

Overall, I have no objections concerning methods or discussion of results. I like the connection between Twitter, Donald Trump, and the COVID-19. There has not been as much research on this topic (Donald Trump) as needed. I expect that data will be available as the authors stated. However, the authors need to mention more literature to strengthen their manuscript. I do not think that it should be very difficult. After that, I would recommend the manuscript for publication.

6. PLOS authors have the option to publish the peer review history of their article (what does this mean?). If published, this will include your full peer review and any attached files.

Reviewer #1: No

Reviewer #2: No

---

## [Author Response · Author response to Decision Letter 0]

20 Apr 2021

Please find the responses to each suggestion/comment by the reviewers below and in the track changes version of the text. We were able to fully respond to all comments and make all of the requested changes and additions. Additionally, we formatted the manuscript according to PLOS ONE guidelines and are depositing the raw dataset in Dryad (https://datadryad.org/stash) where it will be fully available on publication of the manuscript (if accepted). The ethics statement now appears in the Methods section of the manuscript.

Details of response to reviewers: 

Reviewer #1: 

Paragraph 3, line 98 is really something that should be in the discussion and not in the intro.

-This sentence was moved to the conclusion.

Why were those specific 7 words chosen. Can you please explain why these were chosen as opposed to pandemic/lockdown/etc?

-This explanation was added Lines 107-112

From May to July 2020 our research group conducted a preliminary examination of six mainstream news media websites, multiple state public health websites and articles/essays by six prominent science writers as initial research for a larger project to examine communication of scientific and public health information during the COVID19 pandemic. Drawing on the preliminary data from May through July we identified seven keywords to sample narratives about the COVID19 pandemic on Twitter.

Has the AIC been previously derived or is this the creation of the authors. Itʼs logical but this is this measure truly valid?

-The AIC is the authors' creation. We developed this metric to assess the relative impact of individual nodes across successively sampled events. This metric is important because given large participant pools, some individuals may miss some events. As such comparing the mean centrality measures of each individual for the events at which they are present is not enough alone. We needed to scale those means to reflect the relative frequency of each node's presence at the successive events. We have not seen a metric like this employed in the SNA literature, and therefore developed it for our data analysis. We added clarifying text to this effect lines 146-152. 

Line 166 there is a random “s” there.

-Removed

Line 243 – I would use another word other than “ridiculing”. Although likely his intention its presence is more of an opinion than staying with the facts.

-Changed to “mocking” as this is the term that was used across nearly all journalistic coverage in regard to Trump’s tweets and statements regarding masks and Biden between Sept. 22 and October 2, 2020. 

With your questions in the last paragraph, can you suggest some answers/solutions/future directions for research? How can we answer those questions before intervening as suggested.

-Solutions suggested and conclusion revised substantially- see lines 365-385.

Reviewer #2: 

The manuscript addresses an important and relevant topic. It is well-done research. However, I would recommend some minor improvements before publication.

-In the manuscript, there should be more relevant references. In total, there are only 26 references including non-academic sources. The COVID-19 pandemic has been quite studied in social science research. Therefore, authors should add some references to strengthen their manuscript. 

-All articles suggested by Reviewer #2 were added to the manuscript along with two others. 

I would also recommend mentioning research regarding adoption/using Twitter by political actors. In this way, the authors can emphasize the importance of social science research on Twitter. 

-This was added to the manuscript along with the citations (see conclusion)

Regarding methods--The methods are well developed. However, the authors should explain why they chose the period from "July 25th, 2020 to November 15th, 2020" and not a different one. Is it somehow connected with the US general election?

-This was clarified with added text lines 121-122: The period of July 25th to November 15th was selected so that the Twitter project data coincided with concurrent elements of the larger project on tracking COVID19 information.

We hope that the revisions are acceptable, and that the manuscript meets with your approval for publication. Please do not hesitate to inform us of any additional revisions as needed.

---

## [Editor Report · Decision Letter 1]

22 Apr 2021

Social media and public perception as core aspect of public health: the cautionary case of @realdonaldtrump and COVD19

PONE-D-21-05624R1

Dear Dr. Fuentes,

We’re pleased to inform you that your manuscript has been judged scientifically suitable for publication and will be formally accepted for publication once it meets all outstanding technical requirements.

Kind regards,

Siân E Halcrow, Ph.D.

Academic Editor

PLOS ONE
---

## [Editor Report · Acceptance letter]

28 Apr 2021

PONE-D-21-05624R1 

Social media and public perception as core aspect of public health: the cautionary case of @realdonaldtrump and COVD19 

Dear Dr. Fuentes:

I'm pleased to inform you that your manuscript has been deemed suitable for publication in PLOS ONE. Congratulations! Your manuscript is now with our production department. 

Kind regards, 

on behalf of

Dr Siân E Halcrow 

Academic Editor

PLOS ONE